# Predictors of Complete Pathological Response with Chemoimmunotherapy in Triple-Negative Breast Cancer: A Meta-Analysis

Arya Mariam Roy [1,*,†], Supritha Chintamaneni [2,†], Sabah Alaklabi [3], Hassan Awada [1], Kristopher Attwood [4] and Shipra Gandhi [1]

1  Division of Hematology and Oncology, Department of Medicine, Roswell Park Comprehensive Cancer Center, Buffalo, NY 14203, USA; hassan.awada@roswellpark.org (H.A.); shipra.gandhi@roswellpark.org (S.G.)
2  Department of General Medicine, Jagadguru Sri Shivarathreeshwara Medical College, Mysore 570015, India; suprithachintamaneni@gmail.com
3  Division of Medical Oncology, Cancer Center of Excellence, King Faisal Specialist Hospital and Research Center, Riyadh 11211, Saudi Arabia; sfalaklabi@kfshrc.edu.sa
4  Department of Biostatistics and Bioinformatics, Roswell Park Comprehensive Cancer Center, Buffalo, NY 14203, USA; kristopher.attwood@roswellpark.org
*  Correspondence: arya.roy@roswellpark.org; Tel.: +1-925-309-9307
†  These authors contributed equally to this work.

**Simple Summary:** We conducted a meta-analysis to understand the impact of adding immunotherapy to chemotherapy in treating patients with triple-negative breast cancer (TNBC). Combining immunotherapy with chemotherapy increased the chance of attaining complete pathological response in TNBC patients, regardless of their PD-L1 status. The combined therapy worked better for patients with better performance status and positive lymph nodes. Identifying patients who would respond well to this approach can improve treatment decisions and patient outcomes. However, more research is needed to identify biomarkers through which to better understand response and resistance to this therapy.

**Abstract:** Background: Multiple randomized controlled trials (RCTs) have investigated the impact of adding checkpoint inhibitors to neoadjuvant chemotherapy for triple-negative breast cancer (TNBC) patients. However, there is a lack of biomarkers that can help identify patients who would benefit from combination therapy. Our research identifies response predictors and assesses the effectiveness of adding immunotherapy to neoadjuvant chemotherapy for TNBC patients. Methods: We identified eligible RCTs by searching PubMed, Cochrane CENTRAL, Embase, and oncological meetings. For this meta-analysis, we obtained odds ratios using the standard random effects model. To assess the heterogeneity of the study outcomes, the I2 statistic was obtained. Potential bias was assessed using a funnel plot and the corresponding Egger's test. Results: In total, 1637 patients with TNBC were included from five RCTs. Neoadjuvant chemoimmunotherapy significantly improved pCR when compared to neoadjuvant chemotherapy alone. In the subgroup analysis, neoadjuvant chemoimmunotherapy showed higher pCR rates in both Programmed death-ligand 1 (PD-L1)-positive and PD-L1-negative TNBC patients. An Eastern Cooperative Oncology Group (ECOG) performance score (PS) of 0 correlated with increased pCRs (OR = 1.9, $p < 0.001$) in neoadjuvant chemoimmunotherapy vs. neoadjuvant chemotherapy, but no benefit was observed for patients with ECOG PS 1. Nodal positivity was significantly associated with pCR (OR = 2.52, $p < 0.001$), while neoadjuvant chemoimmunotherapy did not benefit patients with negative lymph nodes. Conclusions: Checkpoint inhibition and neoadjuvant chemotherapy significantly increased pCRs in TNBC patients, regardless of their PDL-1 status. Additional checkpoint inhibitors improved pCR rates, mainly for patients with ECOG PS 0 and lymph node-positive disease.

**Keywords:** breast cancer; neoadjuvant chemotherapy; immunotherapy; triple-negative breast cancer; checkpoint inhibitors

## 1. Introduction

Triple-negative breast cancer (TNBC) is an aggressive subtype, characterized by the absence of hormone (estrogen and progesterone) receptors and the human epidermal growth factor receptor 2 (HER2) receptor. This lack of receptors limits the effectiveness of targeted therapies commonly used for other breast cancer subtypes. TNBC has a high rate of recurrence and metastasis, leading to a poorer prognosis than that for different subtypes. It predominantly affects young females, especially those with BRCA mutations [1]. Chemotherapy remains the primary treatment option for TNBC, due to the absence of targeted therapies [2].

One significant measure used to assess treatment response in TNBC patients is complete pathological response (pCR), which indicates the absence of detectable cancer cells in the breast and lymph nodes after neoadjuvant chemotherapy. The pCR rate with neoadjuvant chemotherapy ranges from 35% to 45% [3]. Attaining pCR is a strong prognostic indicator, reducing the risk of cancer recurrence and improving long-term survival outcomes compared to non-pCR cases. Moreover, pCR has emerged as a predictive marker of response to neoadjuvant chemotherapy in TNBC patients. The absence of residual disease after neoadjuvant chemotherapy signifies successful tumor eradication, thus reducing the risk of local recurrence and distant metastasis. Achieving pCR also helps in tailoring subsequent treatment strategies. In selected cases, neoadjuvant chemotherapy may allow for less invasive surgery and mastectomy, leading to better cosmetic outcomes and improved quality of life. Therapy can be de-escalated for patients who achieve pCR by avoiding further systemic therapies in selected cases. This de-escalation approach can minimize unnecessary treatment-related toxicities [4].

Clinical trials have investigated the impact of adding immune checkpoint inhibitors (ICIs) to neoadjuvant chemotherapy in TNBC patients [5–9]. TNBC tumors have been found to contain higher levels of tumor-infiltrating lymphocytes (TILs), indicating a potential immune response against the tumor. Studies have shown that TNBC tumors with higher levels of immune cell infiltration, particularly TILs, are more likely to respond to immune checkpoint inhibitors [10]. Pembrolizumab, a programmed death (PD-1) inhibitor, has been approved for use with neoadjuvant chemotherapy in TNBC patients, based on the clinical benefit observed in Keynote-522. With the addition of immunotherapy to the neoadjuvant chemotherapy regimen, the pCR rate has increased to 65% [6]. However, ICI treatments can lead to side effects that vary in severity and affect multiple organ systems. Due to the increased toxicities associated with ICIs, it is essential to identify patients who are most likely to benefit from this treatment without exposing everyone to the potential side effects.

Currently, there is a lack of biomarkers that can help identify patients who would benefit from neoadjuvant chemoimmunotherapy and aid in de-escalating systemic therapy in selected early-stage TNBC patients [8]. Ongoing clinical trials are investigating targeted therapies such as PI3K/AKT/mTOR inhibitors, EGFR-targeted agents, and antiangiogenic agents in TNBC treatment to identify novel avenues for de-escalation [11–13]. Despite the ongoing advancements outlined, TNBC persists as a puzzling medical challenge, defying definitive solutions.

To address this, we conducted a meta-analysis to identify predictors of response to neoadjuvant chemoimmunotherapy and evaluate the clinical efficacy of adding immunotherapy to neoadjuvant chemotherapy in TNBC patients.

## 2. Materials and Methods

This meta-analysis report was prepared in concordance with the PRISMA (Preferred Reporting Items for Systematic Reviews and Meta-Analyses) 2020 guidelines [14]. The systematic review followed PROSPERO's guide. The systematic review was not registered in PROSPERO.

*2.1. Selection Process and Eligibility*

To identify eligible randomized controlled trials (RCTs) for our study, we thoroughly searched PubMed, Cochrane CENTRAL, Embase, and oncological meetings for relevant studies published up to May 2022 using the following Keywords: "breast cancer", "triple negative breast cancer", "immunotherapy", "immune checkpoint inhibitors", "PD-1", "PD-L1", "Pembrolizumab", "Durvalumab", "Atezolizumab", "neoadjuvant chemotherapy", "neoadjuvant chemoimmunotherapy", "phase II trials", "phase III trials", and "randomized controlled trials". An advanced exploration was conducted by combining the abovementioned keywords or phrases with Boolean operators ('AND' and 'OR'). The eligibility criteria included prospective randomized controlled trials (phase II or phase III) regarding early-stage TNBC, an experimental arm with neoadjuvant chemoimmunotherapy, and a control arm with neoadjuvant chemotherapy alone, as well as the availability of pCR results. There were no restrictions regarding publication date, sample size, country of publication, or line of treatment. Only studies that were published in the English language were considered for eligibility. We excluded phase I studies, non-RCTs, retrospective studies, meta-analyses, studies that included advanced/metastatic TNBC patients, or those including patients who received adjuvant chemoimmunotherapy.

*2.2. Data Collection Process*

Two investigators (AMR and SA) independently screened the studies for eligibility, based on the predefined inclusion and exclusion criteria, and reviewed data from eligible studies. Any discrepancies in study selection between the two reviewers were resolved through consensus or consultation with a third reviewer. Data regarding the study name, study type, study phase, patient characteristics, performance status, and clinicopathological details, such as lymph node involvement, stage, tumor grade, histology, PD-L1 status, neoadjuvant chemotherapy, neoadjuvant chemoimmunotherapy, pCR, and event-free survival (EFS), were collected.

*2.3. Statistical Analysis*

Our meta-analysis focused on the pCR outcome, and we calculated odds ratios with 95% confidence intervals using the standard random effects model. To evaluate the heterogeneity of the study outcomes, we obtained the I2 statistic, which quantified the variability attributed to the study's heterogeneity. Potential bias was assessed using a funnel plot and the corresponding Egger's test. These analyses were applied to the overall cohort of studies and within the specific demographic/clinical sub-cohorts. All analyses were conducted in SAS v9.4 at a significance level of 0.05.

## 3. Results

After the initial screening and assessment of eligibility, five RCTs were included in our meta-analysis (Figure 1). In total, 1637 TNBC patients from five RCTs (Mittendorf 2020, Schmid 2020, Nanda 2020, Loibl 2019, Gianni 2019) [5–9] were included in the study. All of the included studies had neoadjuvant chemoimmunotherapy and neoadjuvant chemotherapy treatment arms. The characteristics of the included studies are given in Table 1. Neoadjuvant chemoimmunotherapy treatment resulted in a significantly improved pCR compared to neoadjuvant chemotherapy alone (Pooled OR = 1.79, 95% CI = 1.28–2.49, $p < 0.001$, $I^2 = 59.1$) (Figure 2).

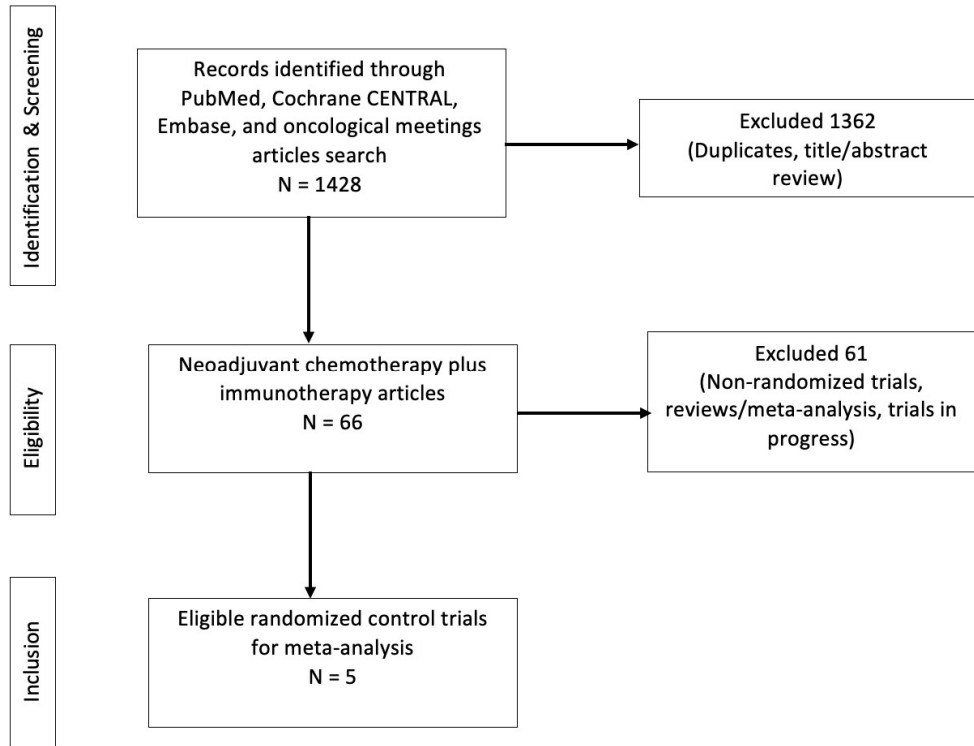

**Figure 1.** A PRISMA flow chart summarizing the literature search and selection of eligible randomized controlled trials.

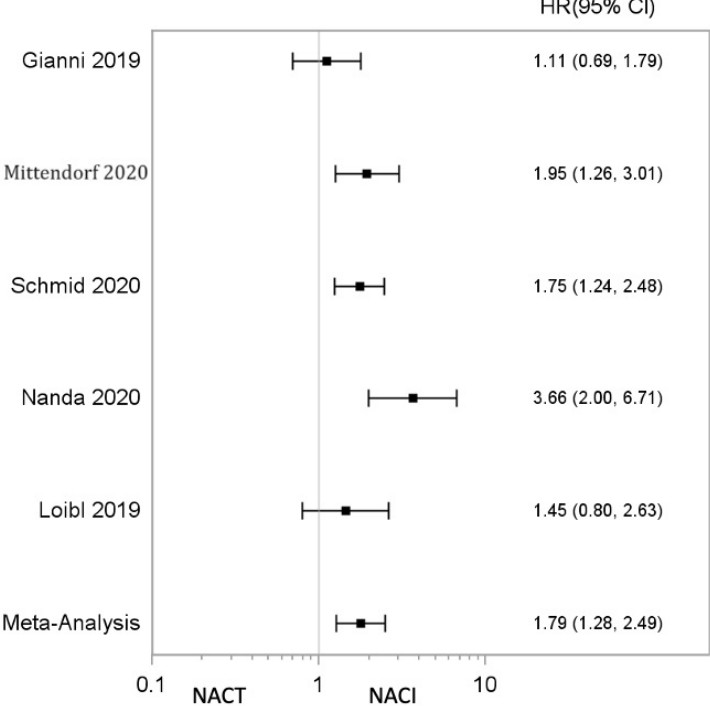

**Figure 2.** A forest plot showing the pCR analysis of the overall cohort shows that neoadjuvant chemoimmunotherapy is better than neoadjuvant chemotherapy alone for the treatment of triple-negative breast cancer [5–9]. NACT = neoadjuvant chemotherapy, NACI = neoadjuvant chemoimmunotherapy.

**Table 1.** Characteristics of included RCTs.

| Clinical Trial | Phase | Sample Size | Intervention vs. Control Arm | Endpoints | Pathological Complete Response (Intervention vs. Control Arm) | Pathological Complete Response (Intervention vs. Control Arm) Based on PD-L1 Status |
|---|---|---|---|---|---|---|
| Gianni 2019 [9]/ NeoTRIP Trial | Phase 2 | 280 | Intervention Arm: Neoadjuvant carboplatin AUC 2 (area under the curve) and nab-paclitaxel 125 mg/m$^2$ i.v. on days 1 and 8, along with atezolizumab 1200 mg i.v. on day 1. Both regimens were given every 3 weeks for eight cycles. Control Arm: Neoadjuvant carboplatin AUC 2 and nab-paclitaxel 125 mg/m$^2$ i.v. on days 1 and 8 without atezolizumab. | Primary endpoint: EFS Secondary endpoint: pCR | 48.6% vs. 44.4%; OR: 1.18; 95% CI = 0.74–1.89; $p$ = 0.48 | PD-L1 +: 59.5% vs. 51.9% PD-L1 −: 33.9% vs. 35.4%. |
| Mittendorf 2020 [5]/IMpassion031 | Phase 3 | 333 | Intervention arm: Chemotherapy plus intravenous atezolizumab (at a dose of 840 mg) every two weeks. Control arm: Chemotherapy plus placebo every 2 weeks Chemotherapy consisted of nab-paclitaxell at 125 mg/m$^2$ weekly for 12 weeks, followed by doxorubicin at 60 mg/m$^2$ and cyclophosphamide at 600 mg/m$^2$ every two weeks for eight weeks. | Co-primary endpoints: pCR in intention-to-treat and PD-L-1-positive patients | 58% vs. 41%; 95% CI = 6–27; $p$ = 0.004 | PD-L1 +: 69% vs. 49% PD-L1 −: 48% vs. 34% |
| Schmid 2020 [6]/Keynote-522 | Phase 3 | 1174 | Intervention Arm: Neoadjuvant therapy with four cycles of pembrolizumab (200 mg) every three weeks. Neoadjuvant therapy included paclitaxel (80 mg/m$^2$ of the body-surface area once weekly) and carboplatin (at a dose based on an area under the concentration–time curve of 5 mg/mL/min once every 3 weeks, or 1.5 mg/mL/min once weekly in the first 12 weeks). An additional four cycles of pembrolizumab were administered. Subsequent treatment with doxorubicin (60 mg/m$^2$) or epirubicin (90 mg/m$^2$) with cyclophosphamide (600 mg/m$^2$ once every 3 weeks in the subsequent 12 weeks). Adjuvant pembrolizumababab was administered every three weeks for up to nine cycles. Control Arm: Neoadjuvant therapy (similar to intervention arm) with four cycles of placebo every three weeks. Adjuvant placebo was administered every three weeks for up to nine cycles. | Co-primary endpoints: pCR at the time of definitive surgery and EFS in the intention-to-treat population | 64.8% vs. 51.2%; 95% CI: 5.4 to 21.8; $p$ < 0.001 | PD-L1 +: 68.9% vs. 54.9% PD-L1 −: 45.3% vs. 30.3% |
| Nanda 2020 [7]/I-SPY2 Trial | Phase 2 | 250 | Intervention arm: Pembrolizumab (200 mg i.v. pembrolizumab every 3 weeks for 4 cycles) to standard neoadjuvant chemotherapy. Control arm: Standard neoadjuvant chemotherapy alone. Neoadjuvant chemotherapy in both arms: 80 mg/m$^2$ i.v. paclitaxel weekly for 12 weeks, followed by 4 cycles of 60 mg/m$^2$ doxorubicin plus 600 mg/m$^2$ i.v. cyclophosphamide every 2 to 3 weeks. | Primary endpoint: pCR Secondary endpoints: RCB, 3-year EFS; distant recurrence-free survival | 60% vs. 22% (TNBC); 95% CI: 44 to 75 | Not available |

**Table 1.** *Cont.*

| Clinical Trial | Phase | Sample Size | Intervention vs. Control Arm | Endpoints | Pathological Complete Response (Intervention vs. Control Arm) | Pathological Complete Response (Intervention vs. Control Arm) Based on PD-L1 Status |
|---|---|---|---|---|---|---|
| Loibl 2019 [8]/ GeparNuevo Trial | Phase 2 | 174 | Intervention arm: ** Durvalumab every 4 weeks with neoadjuvant chemotherapy. Control arm: Placebo with neoadjuvant chemotherapy. Neoadjuvant chemotherapy in both arms: nab-paclitaxenab-paclitaxel weekly for 12 weeks), followed by dose-dense epirubicin 90 mg/m$^2$, and cyclophosphamide 600 mg/m$^2$. | Primary endpoint: pCR | 53.4% (95% CI = 42.5–61.4%) vs. 44.2% (95% CI = 33.5–55.3%; unadjusted continuity corrected χ$^2$$p$ = 0.287 Window phase: 61.0% vs. 41.4%; OR = 2.22; 95% CI = 1.06–4.64; $p$ = 0.035 | PD-L1 +: 58% vs. 50.7% PD-L1 −: 44.4% vs. 18.2% |

Table 1 demonstrates the characteristics of the included studies. pCR: pathological complete response, RCB = residual cancer burden, EFS = event-free survival, CI = confidence interval, OR = odds ratio, mg = milligram, m$^2$ = meters squared, i.v. = intravenous, mL = milliliter, min = minute, g = gram, PD-L1 +: programmed Death-Ligand 1-positive patients, PD-L1 −: programmed Death-Ligand 1-negative patients. ** one injection durvalumab 0.75 g i.v./placebo monotherapy 2 weeks before the start of chemotherapy (window-phase), followed by durvalumab 1.5 g i.v./placebo every 4 weeks.

### 3.1. Factors Affecting Pathological Complete Response

In the subgroup analysis, neoadjuvant chemoimmunotherapy was associated with a higher pCR rate in both PD-L1-positive (OR = 1.66, 95% CI: 1.26–2.17, $p < 0.001$, I statistics 0) (Figure 3) and PD-L1-negative TNBC patients (OR = 1.55, 95% CI: 1.03–2.33, $p = 0.034$, I statistic 0). Only two RCTs studied the impact of the Eastern Cooperative Oncology Group (ECOG) performance score (PS) on the pCR. Mittendorf 2020 reported that, among patients with ECOG PS 0, the pCR rates were 58% vs. 43% with NACI and NACT, respectively. Among patients with ECOG 1, the pCR rates were 63% vs. 21% with NACI and NACT, respectively. Schmid 2020 showed that patients with ECOG 0 had higher pCR rates compared to those with ECOG 1 (NACI vs. NACT: ECOG 0—65.5% vs. 49.1%, ECOG 1—61.6% vs. 64.3%). Our meta-analysis showed that ECOG 0 was associated with increased pCR (OR = 1.90, 95% CI: 1.42–2.53, $p < 0.001$, I statistic 0) with neoadjuvant chemoimmunotherapy vs. neoadjuvant chemotherapy (Figure 4). However, patients with ECOG PS 1 (OR = 2.36, 95% CI: 0.72–7.72, $p = 0.155$) did not derive benefit from neoadjuvant chemoimmunotherapy vs. neoadjuvant chemotherapy. The relationship of pCR with lymph node status was assessed in two trials. Mittendorf 2020 revealed that, for patients with negative regional lymph nodes, the pCR rates with NACI vs. NACT were 58% vs. 49%, respectively, and for patients with positive regional lymph nodes, the pCR rates with NACI vs. NACT were 57% vs. 31%. The Schmid 2020 study found that the pCR rates with NACI vs. NACT in the lymph node-positive group were 64.8% vs. 44.1%, and those of the lymph node-negative group were 64.9% vs. 58.6% respectively. Our meta-analysis showed that pCR was increased in those with positive lymph nodes (OR = 2.52, 95% CI: 1.69–3.77, $p < 0.001$), while there was no benefit with neoadjuvant chemoimmunotherapy in patients with negative lymph nodes (OR = 1.36, 95% CI: 0.94–1.97, $p = 0.103$) (Figure 5). These results appear to be consistent across the different demographic/clinical sub-groups. However, there was a statistically significant difference in the effect of immunotherapy (when added to chemotherapy) between node-positive and -negative TNBC patients, in which a greater benefit (relative to chemotherapy alone) was observed in the node-positive patients. We found a significant association between nodal positivity and pCR (OR $p = 0.023$).

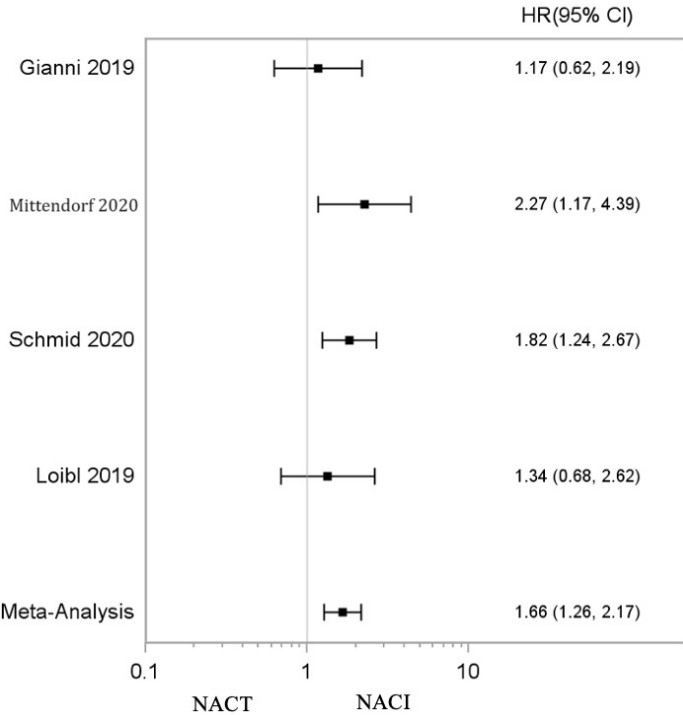

**Figure 3.** A forest plot showing the impact on pCR in patients with PDL-1 positivity with triple-negative breast cancer [5–9]. NACT = neoadjuvant chemotherapy, NACI = neoadjuvant chemoimmunotherapy.

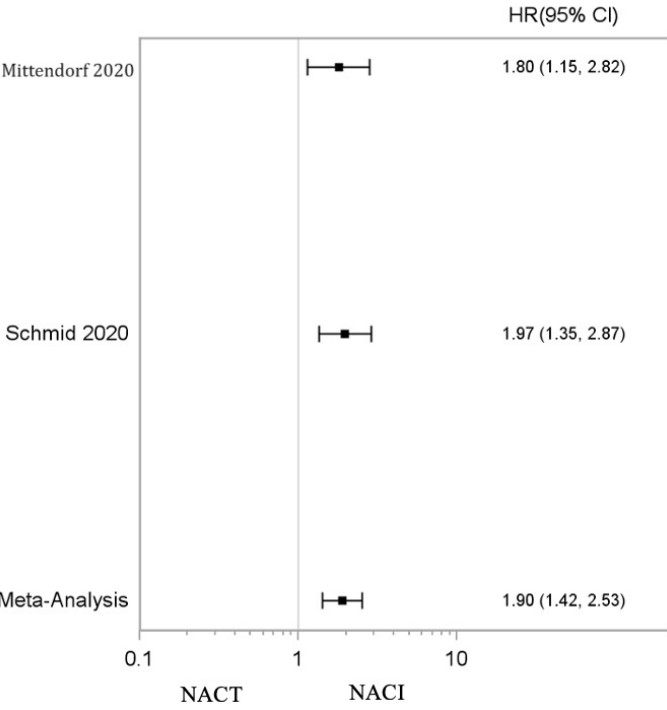

**Figure 4.** A forest plot showing the impact on pCR in patients with ECOG 0 performance status with triple-negative breast cancer [5,6]. NACT = neoadjuvant chemotherapy, NACI = neoadjuvant chemoimmunotherapy.

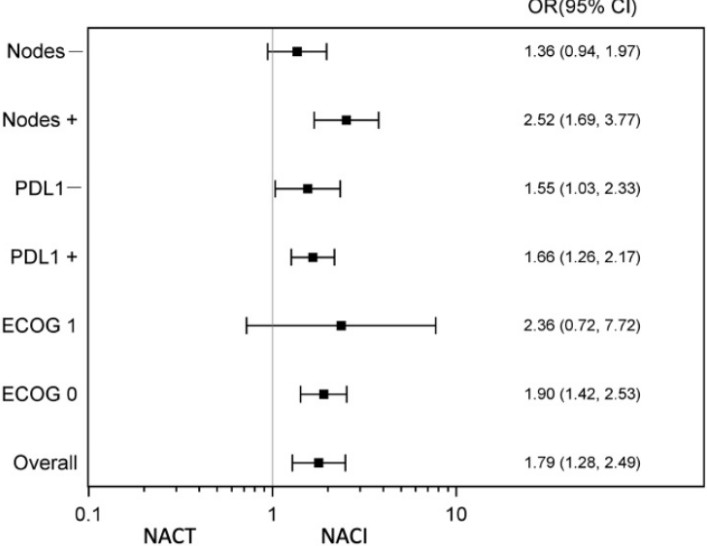

**Figure 5.** A forest plot showing the subset analysis of factors predicting pCR. NACT = neoadjuvant chemotherapy, NACI = neoadjuvant chemoimmunotherapy.

*3.2. Impact of Addition of Neoadjuvant Immunotherapy on Survival Outcomes*

The majority of the included studies did not report an analysis of clinical outcomes based on the response to neoadjuvant treatments, which is our topic of interest. Some of the included studies did not have data on clinical outcomes after neoadjuvant treatment, such as EFS and OS. From the meta-analysis of the trials that reported results of EFS (Keynote-522 and IMPassion031), we found that neoadjuvant chemoimmunotherapy improves EFS (Pooled HR = 0.66, 95% CI: 0.48–0.92, *p* = 0.015, I statistic 0) (Figure 6). There was no evidence of publication bias in our study (Figure 7).

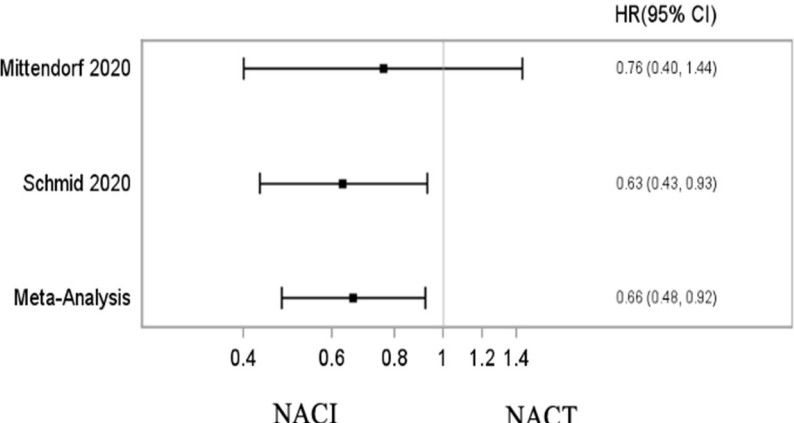

**Figure 6.** Forest plot showing event-free survival, a benefit of the addition of immunotherapy to neoadjuvant chemotherapy regimens [5,6]. NACT = neoadjuvant chemotherapy, NACI = neoadjuvant chemoimmunotherapy.

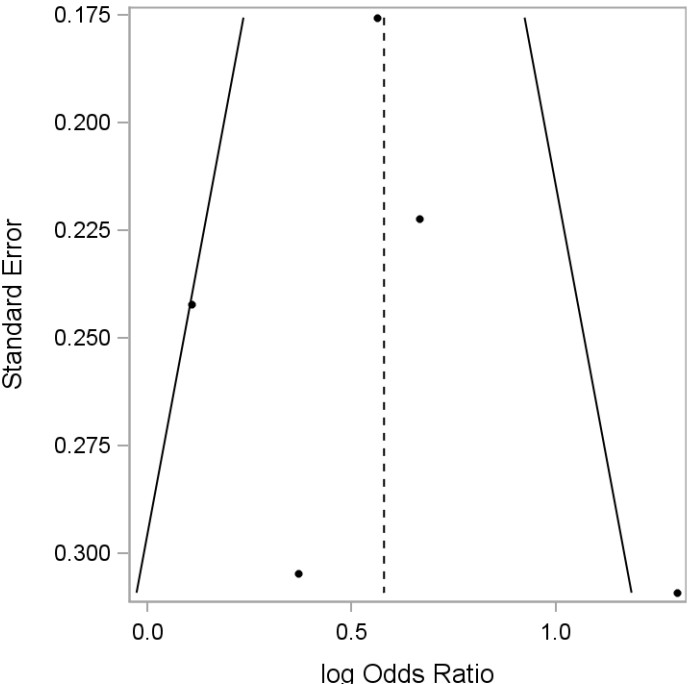

**Figure 7.** The funnel plot shows no publication bias.

## 4. Discussion

Neoadjuvant chemotherapy has been the standard treatment approach for TNBC, aiming to downstage the tumor and improve surgical outcomes [15]. Recently, the standard of care for early-stage TNBC patients has been changed to neoadjuvant chemoimmunotherapy, after the Keynote-522 trial showed a survival benefit in adding ICIs to neoadjuvant chemotherapy regimens [5]. Since pCR is a surrogate marker for improved survival in TNBC patients [16], several studies have evaluated treatment combinations to identify regimens that can increase pCR. However, not all patients achieve pCR following neoadjuvant chemotherapy, and identifying predictors of pCR can help guide treatment decisions and improve treatment efficacy, enhancing patients' quality of life without increased toxicities. Although Keynote-522 improved EFS with the addition of pembrolizumab with neoadjuvant chemotherapy, the results of similar studies have not been consistent [4–8,17,18]. Therefore, we conducted a meta-analysis, which included a large cohort of 1637 TNBC patients from five randomized controlled trials [4–8] to evaluate the impact of the addition of

ICIs with neoadjuvant chemotherapy and predictors of pCR in this population. In line with prior studies, our results demonstrated that incorporating ICIs into neoadjuvant chemotherapy can significantly enhance the chances of achieving pCR in TNBC patients [19]. The data from the included studies are not mature enough to analyze the overall survival of the patients.

Several studies have shown that the achievement of pCR after neoadjuvant treatment is associated with improved survival outcomes, such as EFS and overall survival (OS) [16]. The attainment of pCR with neoadjuvant treatment depends on tumor biology. Studies show that HER2-positive breast cancer and TNBC patients achieve higher rates of pCR; however, hormone receptor-positive breast cancer tends to have a lower pCR to neoadjuvant treatments [20]. The presence of residual disease after neoadjuvant treatment is associated with higher relapse rates in TNBC patients [21]. Although we utilize pCR as an endpoint in clinical trials utilizing neoadjuvant systemic treatments, a longer follow-up time through which to assess EFS and OS is necessary, as pCR might not necessarily translate to better clinical response in some patients, especially in certain racial groups and in certain breast cancer subtypes [22,23].

Predictive biomarkers are important for distinguishing responders and non-responders to neoadjuvant chemoimmunotherapy. Several biomarkers, such as the expression of PD-L1, the tumor mutation burden (TMB), the neoantigen load, TILs, circulating tumor DNA, and the gut microbiome, are linked with response and resistance to immunotherapy [24]. Our subgroup analysis revealed that both PD-L1-positive and PD-L1-negative TNBC patients benefited from neoadjuvant chemoimmunotherapy, which suggests that the pCR is independent of the PD-L1 status of the patients [25]. We also assessed the impact of the patients' performance score (PSs) on pCR rates. We used ECOG scoring for assessing the PS, as it is widely used in the included clinical trials. Patients with better PSs, as denoted with an ECOG of 0, were more likely to achieve pCR with neoadjuvant chemoimmunotherapy than those with limited PSs. This suggests that patients with better performance status may benefit more significantly from adding ICIs to neoadjuvant chemotherapy [26].

Our study also demonstrated a significant association between nodal positivity and pCR rates in TNBC patients. Our finding suggests that TNBC patients with lymph node (LN) involvement may derive more pCR benefits from neoadjuvant chemoimmunotherapy than those without LN involvement. This could be attributed to multiple factors. Positive LNs often reflect a more extensive disease burden and a higher tumor stage. These larger tumors may be more responsive to neoadjuvant treatment, including chemotherapy and immunotherapy, resulting in a higher probability of complete tumor regression. Positive LNs may indicate a more aggressive tumor biology, with a higher proliferative rate. Highly proliferative tumors tend to be more sensitive to cytotoxic therapies, including chemotherapy, which increases the likelihood of achieving pCR [27,28]. The role of the LNs in the immune response against cancer cells, acting as sites for immune cell activation and antigen presentation, could be another reason behind this improved pCR with neoadjuvant chemoimmunotherapy. LNs serve as immunological hubs where immune cells interact with cancer cells and initiate an antitumor immune response. The infiltration of cancer cells into the lymph nodes may trigger immune activation, making the tumor more susceptible to immunotherapeutic interventions [29]. Identifying nodal involvement as a predictive factor for pCR can help clinicians in treatment decision making and patient management. Patients with positive lymph nodes may be prioritized for neoadjuvant therapy regimens that include immunotherapy, as they are more likely to benefit from this treatment approach. Conversely, patients with negative LNs might be spared from the potential toxicities associated with immunotherapy, without sacrificing treatment efficacy [27,30,31]. While the presence or absence of LN involvement is a crucial predictor, recent research indicates that more complex immune-related genetic signatures, neoantigens, and mutational burdens may improve our ability to predict pCR and the overall response to breast cancer treatments [31,32].

Several real-world studies have shown that race can be a predictor for pCR in several malignancies [23,33,34]. Zhao et al. have shown that Black patients have lower odds of achieving pCR in hormone receptor-negative HER2-positive breast cancer [33]. A study from our group showed similar findings. We found that Black patients have higher refractory disease (a pathological stage higher than or equal to the clinical stage) with neoadjuvant treatments compared to all other racial groups in TNBC and had a lower percentage of very sensitive disease (pathological complete response) in the HER2-positive breast cancer subtype. Furthermore, we found that Black patients with refractory and sensitive disease (a pathological stage lower than the clinical stage) have higher mortality compared to Whites. Asians were found to have improved mortality in pCR and residual disease groups compared to other races. Some studies have mentioned differences in tumor biology between races, based on the immune microenvironment composition. Black women were found to have significantly higher CD8+ T cell density [35]. Another study showed a higher number of TILs in Asians and Native Hawaiian/Pacific Islanders [36]. This difference in the immune microenvironment can play a role in the responses of different races to immunotherapy [23]. However, most of the trials in our meta-analysis did not have subset analysis based on race, so we could not assess the association of race and pCR/survival in our meta-analysis [34].

Although age at diagnosis has been mentioned as a predictor of the response to neoadjuvant chemotherapy in several malignancies, there are disparities in the results of several studies for breast cancer, and no consensus has been reached. A study by Li et al. demonstrated no significant association between age group and neoadjuvant chemotherapy treatment response. However, they observed that patients aged >50 years who attained pCR after neoadjuvant chemotherapy experienced better survival compared to young patients [37]. A pooled analysis of eight neoadjuvant trials showed that women <40 years of age with breast cancer have a higher likelihood of attaining pCR, and this was more pronounced in HER2-negative breast cancer patients [38]. Another study observed that an age <50 years is an independent predictor of pCR in breast cancer patients [39]. However, subset analysis based on age groups was not reported in the majority of the included RCTs, which limited our ability to analyze the impact of age on pCR with neoadjuvant chemotherapy and neoadjuvant chemoimmunotherapy.

Our study has several strengths and limitations. Our study included all published clinical trials that included neoadjuvant chemoimmunotherapy in the treatment of TNBC. The results of this study were consistent across different demographic and clinical subgroups, indicating the robustness of the findings. Moreover, there was no evidence of publication bias, adding further credibility to the results. Given that our meta-analysis incorporated only prospective RCTs, the limitations associated with retrospective methodologies, such as selection bias, information bias, and confounding, are limited. One of the major limitations is that most of the included RCTs did not consistently report other factors that could predict responses to neoadjuvant chemoimmunotherapy, such as age, race/ethnicity, tumor grade, tumor size, TILs, or tumor mutational burden. Another limitation is that some of the included RCTs did not report survival outcomes, such as EFS, which limited our ability to assess the long-term survival outcomes of patients who received neoadjuvant chemoimmunotherapy. Also, we could not assess the impact of pCR on survival outcomes such as EFS and OS, as these data were not reported in the majority of the included RCTs. Furthermore, the data from most of the involved RCTs are not mature enough to analyze the impact of pCR on overall survival.

Further studies are needed to identify additional possible biomarkers for predicting the response to neoadjuvant chemoimmunotherapy in TNBC patients, aiming to refine treatment protocols, pinpoint biomarkers, and unravel the mechanisms underlying both response and resistance to neoadjuvant chemoimmunotherapy. A recent study has revealed that cellular composition and multicellular spatial organization are major determinants of ICI effectiveness and can be used to predict responses to immunotherapy pre-treatment [40].

Molecular profiling of breast cancer is useful in identifying patients who are most likely to benefit from neoadjuvant chemotherapy and tailoring treatment accordingly.

## 5. Conclusions

In conclusion, this study provides valuable insights into predictors of pCR in TNBC patients treated with neoadjuvant chemotherapy. Adding immune checkpoint inhibitors to neoadjuvant chemotherapy significantly improved pCR rates, irrespective of PD-L1 expression status. Factors such as ECOG PS and LN status also influenced the likelihood of achieving pCR. Identifying potential biomarkers or predictors of response for neoadjuvant chemoimmunotherapy would be beneficial in selecting patients likely to have clinical responses to the treatment, while minimizing the risk of unwanted side effects. Although pCR is a reliable surrogate sign, its ability to translate into longer survival benefits is still crucial to consider, particularly in neoadjuvant chemotherapy and immunotherapy.

**Author Contributions:** Conceptualization, A.M.R. and S.G.; methodology, A.M.R. and S.A.; software, K.A.; validation, A.M.R., S.A. and S.G.; formal analysis, K.A.; data curation, A.M.R., S.A. and H.A.; writing—original draft preparation, A.M.R. and S.C.; writing—review and editing, S.A., H.A. and S.G.; supervision, S.G. All authors have read and agreed to the published version of the manuscript.

**Funding:** This research received no external funding.

**Institutional Review Board Statement:** Ethical review and approval were waived for this study, as the data were obtained from publicly available published studies.

**Informed Consent Statement:** Patient consent was waived, as the data were obtained from publicly available published articles.

**Data Availability Statement:** The data presented in this study are openly available in PubMed, Cochrane CENTRAL, Embase, and oncological meetings.

**Conflicts of Interest:** The authors declare no conflicts of interest.

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
