# Peer review of "Predictors of Complete Pathological Response with Chemoimmunotherapy in Triple-Negative Breast Cancer: A Meta-Analysis"

_onco, doi:10.3390/onco4010001_

Round 1

Reviewer 1 Report

Comments and Suggestions for Authors

1.     Specify the search string used for data curation.

2.       Suggest to refresh data collection time panel to November 2023.

3.       Correct the graphic notations, such as “Mittendorf 20”in Figure2/3/4 to “Mittendorf 2020”。

4.       Add the comparison data of pCR rates of TNBC patients treated with NACI between PD-L1 positive ones and PD-L1 negative ones.

5.       Add the comparison data of pCR rates of TNBC patients treated with NACI between ECO G1 ones and ECO G0 ones.

6.       Add the comparison data of pCR rates of TNBC patients treated with NACI between Nodes positive ones and Nodes negative ones.

Comments on the Quality of English Language

Minor editing of English language required.

Author Response

Dear Reviewer,

Thank you for taking the time to review the manuscript. We appreciate your valuable comments. Provided below are point-to-point responses to your comments.

Comment 1: Specify the search string used for data curation.

Response 1: The search string that we used was: (“breast cancer” OR “triple negative breast cancer”) AND (“immunotherapy” OR “immune checkpoint inhibitors” OR “PD-1” OR “PD-L1” OR “Pembrolizumab” OR “Durvalumab” OR “Atezolizumab” OR “neoadjuvant chemotherapy” OR “neoadjuvant chemoimmunotherapy”) AND (“phase II trials” OR “phase III trials” OR “randomized control trials).

As we mentioned the keywords in the main manuscript, we added a sentence under the “Selection Process and Eligibility’” saying that “An advanced exploration was conducted by combining above-mentioned keyword words or phrases with Boolean operators (‘AND,’ ‘OR’).”

Comment 2: Suggest to refresh data collection time panel to November 2023. 

Response 2: The data collection and analysis for the project were completed earlier. Expanding the search criteria now to include articles until November 2023 would significantly alter the manuscript, requiring a new flow diagram and changes in the results section. We will consider this for future studies, especially when results from ongoing trials become available. Thank you for your understanding.

Comment 3: Correct the graphic notations, such as “Mittendorf 20”in Figure 2/3/4 to “Mittendorf 2020.”

Response 3: Appropriate changes have been made.

Comment 4: Add the comparison data of pCR rates of TNBC patients treated with NACI between PD-L1 positive ones and PD-L1 negative ones.

Response 4: Appropriate information has been added. Please see below.

Comment 5: Add the comparison data of pCR rates of TNBC patients treated with NACI between ECO G1 ones and ECO G0 ones.

Response 5: Appropriate information has been added.

Comment 6: Add the comparison data of pCR rates of TNBC patients treated with NACI between Nodes positive ones and Nodes negative ones.

Response 6: Appropriate information has been added.

Reviewer 2 Report

Comments and Suggestions for Authors

1. Need to describe at least some uncommon abbreviations.

2. Move line 45 to top of page 2.

3. Somewhere this is needed perhaps in the Introduction: "Improvements are happening as we describe here but TNBC remains essentially an unsolved  disease."

4. in Table 1, column information should be left side indexed.

5. figure 4 needs to be reduced in size.

Author Response

Dear Reviewer,

Thank you for taking the time to review the manuscript. We appreciate your valuable comments. Provided below are point-to-point responses to your comments.

Comment 1: Need to describe at least some uncommon abbreviations.

Response 1: We have expanded the abbreviations of NACI and NACT. We find the others to be standard abbreviations. Thank you for your understanding.

Comment 2: Move line 45 to the top of page 2.

Response 2: Appropriate changes have been made.

Comment 3: Somewhere, this is needed, perhaps in the Introduction: "Improvements are happening as we describe here, but TNBC remains essentially an unsolved  disease."

Response 3: Thank you for the suggestion. An appropriate sentence has been added under "Introduction."

Comment 4: In Table 1, column information should be left-side indexed. 
Response 4: Appropriate changes have been made.

Comment 5: Figure 4 needs to be reduced in size. 
Response 5: Appropriate changes have been made.

Reviewer 3 Report

Comments and Suggestions for Authors

It is a meta-analysis to identify predictors of pathological complete response(pCR) after neoadjuvant immunochemotherapy for triple-negative breast cancer. They found that ECOG PS=0 and lymph node-positive disease were the predictors for pCR after neoadjuvant immunochemotherapy. Moreover, they confirmed that checkpoint inhibitors significantly increased pCR in TNBC patients regardless of the PDL-1 status. Nevertheless, I suggest a minor revision before approval.

1.     As the target population was the triple-negative breast cancer patients who received neoadjuvant immunochemotherapy or chemotherapy only, please justify your title to be more precise.

2.     The different ages of patients, regimens of chemotherapy, BRCA1/2, or other genetic information, if any, should also be included in your analysis and discussions.    

3.     Compared with similar articles, not much new knowledge has been drawn from this study, just like some limitations in your discussion.

4.     In line 33, the full name of ‘NACI’ is missing.

5.     Statistical review is warranted.

Author Response

Dear Reviewer,

Thank you for taking the time to review the manuscript. We appreciate your valuable comments. Provided below are point-to-point responses to your comments.

Comment 1: As the target population was the triple-negative breast cancer patients who received neoadjuvant immunochemotherapy or chemotherapy only, please justify your title to be more precise.

Response 1: Appropriate changes to the title have been made.

Comment 2: The different ages of patients, regimens of chemotherapy, BRCA1/2, or other genetic information, if any, should also be included in your analysis and discussions.

Response 2: We tried to do the mentioned analysis. However, these data, especially subset analysis based on the age of the patients, BRCA 1/2, were not given in most of the included RCTs. Also, it was not in the forest plots or supplementary details. Details of the chemotherapy regimen used in all these trials are given in Table 1, including the drug details and their dose and administration. We believe this is unnecessary in the discussion as the data was unavailable in the RCTs. None of the included trials stratified patients based on the BRCA 1/2 status and genetic mutations. Furthermore, this has been mentioned as a limitation in the article. Thank you for your understanding.

Comment 3: Compared with similar articles, not much new knowledge has been drawn from this study, just like some limitations in your discussion.

Response 3: This article has shed light on the necessity of identifying additional predictive biomarkers to customize treatments for patients with TNBC. It further emphasizes the importance of stratifying clinical trials based on all relevant factors impacting chemo-immunotherapy treatment responses and toxicities, including age, sex, race, tumor size, grade, and genetic mutations. We appreciate your understanding.

Comment 4: In line 33, the full name of ‘NACI’ is missing.

Response 4: Appropriate changes have been made.

Comment 5: Statistical review is warranted.

Response 5: Our statistician conducted the analysis outlined in the paper and thoroughly reviewed it internally prior to submission. Thank you for your understanding.